

# Development of the first high-density linkage map in the maize weevil, *Sitophilus zeamais*

Jennifer Fountain Baltzegar[1] and Fred Gould[1,2]

[1] Genetic Engineering and Society Center, North Carolina State University, Raleigh, NC, United States
[2] Department of Entomology and Plant Pathology, North Carolina State University, Raleigh, North Carolina, United States

## ABSTRACT

The maize weevil, *Sitophilus zeamais*, is a worldwide pest that disproportionately affects subsistence farmers in developing countries. Damage from this pest threatens food security in these communities as widely available and effective control methods are lacking. With advances over the last decade in the development of genetic pest management techniques, addressing pest issues at the ecosystem level as opposed to the farm level may be a possibility. However, pest species selected for genetic management techniques require a well-characterized genome and few genomic tools have been developed for *S. zeamais*. Here, we have measured the genome size and developed the first genetic linkage map for this species. The genome size was determined using flow cytometry as 682 Mb and 674 Mb for females and males, respectively. The linkage map contains 11 linkage groups, which correspond to the 10 autosomes and 1 X-chromosome found in the species and it contains 1,121 SNPs. This linkage map will be useful for assembling a complete genome for *S. zeamais*.

# INTRODUCTION

The maize weevil, *Sitophilus zeamais* Motschulsky, is a pest of maize and other cereal crops. The exact origin of the species is unknown but is likely from Asia and its worldwide distribution is thought to have been facilitated through grain trade (*Corrêa et al., 2017*). Despite efforts to control *S. zeamais*, it remains an important economic pest in the developing world. Subsistence farmers, who store grain for months after harvest, are particularly vulnerable to damage from *S. zeamais*. In severe infestations, farmers can lose the majority of their grain or be forced to sell in a less favorable market to prevent complete loss of harvest (*Stephens & Barrett, 2011*). With advances over the last decade in the development of advanced genetic pest management techniques using CRISPR/Cas9, addressing pest issues at the ecosystem level as opposed to the farm level is a possibility (*Esvelt et al., 2014*). Genetic pest management techniques such as male-selecting transgenic lines or gene drives designed to suppress or replace damaging phenotypes in the population could be used to mitigate pest damage. Some of these techniques are currently being developed in several insect pest species, including *Plutella xylostella, Anopheles*

Corresponding author
Jennifer Fountain Baltzegar,
jen_baltzegar@ncsu.edu

*gambiae, Aedes aegypti, Lucillia cuprina*, and others (*Harvey-Samuel et al., 2015*; *Kyrou et al., 2018*; *Anderson et al., 2023*; *Yan et al., 2020*). It is likely that if these techniques succeed in the field, then they will also be considered for development in other pest species. *S. zeamais* may be a targeted species because of its widespread geographic distribution and the threat to food security that it poses to subsistence farmers in developing nations (*Corrêa et al., 2017*; *García-Lara, García-Jaimes & Bergvinson, 2019*; *Baltzegar et al., 2023*). However, for genetic-based technologies to be a feasible option for controlling *S. zeamais*, more species-specific genetic information and tools need to exist.

Few genomic tools are available for use with *S. zeamais* that would allow for quantitative trait loci (QTL) analysis, genome-wide association studies (GWAS), or complex genomic manipulation in the laboratory. The genomic tools that have been developed in this species primarily consist of karyotypes, evolutionary and species-specific diagnostic markers, microsatellite markers, and primer sequences for amplification of specific gene fragments. Several karyotypes have been produced for the species with consensus that *S. zeamais* has 10 pairs of autosomes and a heterogametic sex chromosome with an Xyp sex-determination mechanism (*Smith & Brower, 1974*; *Zhi-Yua, Pei & Guo-Xiong, 1989*; *da Silva et al., 2015*, *2018*). The genome size was reported as 713.5 Mb for females and 709.3 Mb for males with no B chromosomes (*da Silva et al., 2018*). Interestingly, the occurrence of two types of supernumerary B chromosomes have been found in this species and range from 0 to 6 in individuals (*Smith & Brower, 1974*; *da Silva et al., 2015*). The Type I B chromosomes are found in both sexes while the Type II B chromosomes are only found in males. Neither are inherited in a Mendelian fashion (*da Silva et al., 2015*). Random amplification of polymorphic DNA (RAPD), restriction fragment length polymorphism (RFLP), nuclear internal transcribed spacer region (ITS), cytochrome C oxidase subunit I (COI), and cytochrome C oxidase subunit II (COII) PCR primers have been developed to molecularly distinguish *S. zeamais* from *S. oryzae* as these two species are almost morphometrically indistinguishable except for the genitalia and are challenging to identify without a microscope (*Hidayat, Phillips & Ffrench-Constant, 1996*; *Peng et al., 2003*; *Corrêa et al., 2013*, *2014*). Recently, effort to elucidate the evolutionary history of these two species has been conducted using mitochondrial COI and COII markers and the nuclear ITS marker (*Corrêa et al., 2017*). The authors found that *S. zeamais* and *S. oryzea* diverged approximately 8.7 million years before present. *Ndiaye & Sembène (2018)* used cytochrome B and COI markers to assess the population genetic structure of *S. zeamais* populations in West Africa. Although this study identified several common haplotypes, they were not able to identify any phylogeographic patterns associated with their distribution. Similarly, *Corrêa et al. (2017)* used nine polymorphic microsatellite markers and found very little phylogeographic structure when comparing *S. zeamais* populations throughout the world. This may indicate a recent geographic expansion of the species, possibly facilitated through global grain trade.

Insecticide resistance is another important area of research for this species and resistance to dichlorodiphenyltrichloroethane (DDT) has been reported since 1970 (*Guedes et al., 1995*). Resistance to DDT and cross-resistance to pyrethroids was confirmed by *Guedes et al. (1995)* and although several studies have reported insecticide resistance in

various strains of *S. zeamais*, few studies have investigated the molecular basis of resistance. One study sequenced the region of the voltage-gated sodium channel (*VGSC*) gene that encodes domain II of the protein (*Araújo et al., 2011*). They identified the single nucleotide polymorphism (SNP) T929I that has been associated with phenotypic resistance to pyrethroids in other arthropod species and developed a TaqMan Assay to genotype individual weevils (*Schuler et al., 1998*; *Lee et al., 2000*). The T929I SNP was identified in pyrethroid resistant lab populations, but it was not found at high frequencies in field-collected samples (*Araújo et al., 2011*; *Haddi et al., 2018*). Additional PCR primer sequences have been developed for genes of interest including the gene family encoding cysteine proteinases, aldolase, prolactin receptor, and interleukin-1β (*Matsumoto et al., 1997*; *Peng et al., 2003*).

Two recent sets of genomic studies have produced transcriptomes and followed up with functional analyses. *Huang et al. (2018)* compared global transcriptomes of *S. zeamais* that had either been exposed to Terpinen-4-ol, an active ingredient in tea tree essential oil used as an insecticide, or those that had not been exposed. Then they conducted a functional analysis to identify specific cytochrome p450s involved in the susceptibility of *S. zeamais* to Terpinen-4-ol (*Huang et al., 2020*). *Tang et al. (2019a)* produced an antennal transcriptome and identified 41 candidate odorant binding protein genes. They followed up by utilizing quantitative real-time PCR to compare the expression of odorant receptors in seven different tissues (*Tang et al., 2019b*). Cytochrome P450s play important roles in insecticide resistance to chemicals, such as the Terpinen-4-ol, while odorant binding proteins and odorant receptors are important because they are used by the insect to identify food sources and potential mates.

Over the last two decades, researchers have put effort into developing basic molecular tools for *S. zeamais*, but more genomic-level resources are needed if novel, genetic-based pest management techniques are to be considered for mitigating damage from this pest. In the age of next generation sequencing and bioinformatics, researchers are left at a disadvantage if they do not possess basic genome-wide polymorphism data. Therefore, the goals of this research are to (1) confirm the genome size of *S. zeamais* as this is a necessary parameter required for next generation sequencing, and (2) develop the first genetic linkage map for *S. zeamais*, which will provide a set of genome-wide SNPs. The high-density radtag linkage map from an $F_2$ population will also be useful during a future genome assembly or quantitative and functional genetic studies that require genome-wide sequence information.

## MATERIALS AND METHODS

### Laboratory colonies for genetic linkage map

*Sitophilus zeamais* colonies were acquired from USDA laboratories in Gainesville, Florida and Manhattan, Kansas. The origin of the colony obtained from Florida is unknown. The origin of the colony obtained from Kansas was the Stored Products Insects Lab, Tifton, GA prior to 1961. An $F_2$ population was generated by mating two siblings from an $F_1$ cross of a single male from the Kansas colony bred to a single female from the Florida colony. Genomic DNA was isolated from the head and thorax of each $F_2$ offspring and the parents

and grandparents using the Qiagen DNeasy kit (Qiagen, Inc., Valencia, CA, USA) with an RNase A treatment following an overnight tissue lysis incubation at 55 °C. The breeding individuals and individuals from the $F_2$ generation were used to generate restriction-site associated DNA (RAD) loci following the double-digest restriction-site associated sequencing (ddRadSeq) procedure outlined below.

## Genome size estimation

The genome size of *S. zeamais* was confirmed following the protocol by *Hare & Johnston (2011)*. In brief, neural cells from the heads of one *S. zeamais* and one *Drosophila melanogaster* were isolated together and stained with propidium iodide. Relative fluorescence of the stained neural cells was measured on a Becton Dickinson LSRII (BD Biosciences, San Jose, CA, USA) flow cytometer at the Flow Cytometry Core Facility, NC State University. In total, five male *S. zeamais* and five female *S. zeamais* individuals were used as biological replicates. Female and male genome sizes were estimated separately to account for the differences expected from the heterogametic XY chromosomes in this species. The following formula was used to determine the genome size of *S. zeamais* samples, where Sz = *S. zeamais* and Dm = *D. melanogaster* and the genome size for *D. melanogaster* is 175 Mb (*Hare & Johnston, 2011*; *Bennett et al., 2003*). Presence and number of B chromosomes in these individuals was not assessed.

$$Genome\ Size\ _{Sz} = Genome\ Size\ _{Dm} \times \frac{Fluorescence_{Sz}}{Fluorescence_{Dm}}$$

## Double-digest restriction-site associated sequencing (ddRadSeq)

Following procedures outlined in *Fritz et al. (2016)*, genomic DNA from individual weevils was digested with *Msp1* and *EcoRI-HF* restriction enzymes and Cutsmart Buffer (New England Biolabs, Inc., Ipswich, MA, USA) at 37 °C for 3 h. Adapters were ligated onto the sticky ends of the digested material using T4 DNA Ligase (New England Biolabs, Inc., Ipswich, MA, USA) at 22 °C for 1 h, followed by 65 °C for 30 min. Proper ligation was confirmed *via* PCR. 5 µl of each sample was pooled into one of five indexed libraries and PCR purified using a QIAquick PCR Purification Kit (Qiagen, Inc., Valencia, CA, USA). Prior to preparing individuals for next generation sequencing, we had to first determine the best size selection window. Following the procedure outlined in *Peterson et al. (2012)*, restriction digests for each restriction enzyme used (MspI and EcoRI-HF) were performed independently on gDNA isolated from an individual *S. zeamais* reared in the laboratory colony. The digest products were then analyzed on a 2100 Bioanalyzer (Agilent Technologies, Inc., Santa Clara, CA, USA) using a high sensitivity DNA assay and the electropherograms were examined to determine the proportion of genome that would be sampled in a given fragment size window. Based on these results, a size-selection window of 400–800 bp was chosen with a target fragment size of 600 bp. Each library was then size selected using these criteria on a Blue Pippin Prep (Sage Sciences, Inc., Beverly, MA, USA) electrophoresis platform in the Genomic Sciences Laboratory at NC State University (GSL). Following size-selection, libraries were PCR amplified using primers specific to the

corresponding Illumina index. Then, PCR product was pooled and purified two times with a QIAquick PCR Purification Kit (Qiagen, Inc., Valencia, CA, USA) and DNA quantity was measured on an Agilent 2200 TapeStation (Agilent Technologies, Inc., Santa Clara, CA, USA) using a Hi-Sensitivity D1000 tape in the GSL. Final library concentrations were diluted to 4 nM and submitted for sequencing on the Illumina Hi-Seq 2500 (Illumina, San Diego, CA, USA), 125 bp SE in the GSL. Two independent samples with separate barcodes and indices were prepared and sequenced for each parent and grandparent to ensure sufficient coverage for catalog development.

### Initial bioinformatic analysis

Illumina sequences were demultiplexed by index and quality checked by FastQC prior to filtering (*Andrews, 2010*). Sequences were then trimmed to remove over-represented sequences by implementing Trimmomatic v 0.32 (*Bolger, Lohse & Usadel, 2014*). Trimming was confirmed by FastQC and a MultiQC report containing FastQC data for all indices was generated (*Ewels et al., 2016*). Then, individual components of the Stacks program were run as follows (*Catchen et al., 2013*). *Process_radtags* was first implemented to screen raw reads for quality, demultiplexed by barcode, and truncated to 115 bp. Then, *ustacks* ($-M = 5$, $-m = 3$) was run to identify putative loci, or stacks, from the raw processed radtags. Next, a catalog of loci was built from the concatenated files of the F1 parents using *cstacks* ($-n = 5$). The stacks of all individuals were then compared to the catalog in *sstacks*. Finally, *genotypes* ($-t = cp$) was run to produce the output for OneMap.

### Genetic linkage map construction

The initial output produced by Stacks for use in OneMap was further filtered by requiring Mendelian inheritance and ~96% (100 out of 104) of individuals to be genotyped for each marker. Mendelian inheritance was confirmed with a Chi-Square test. A genetic linkage map was constructed in R (*R Core Team, 2020*) with the *onemap* package using the Kosambi mapping function and the Record function to order the markers (*Margarido, Souza & Garcia, 2007*). An SVG image file containing a visual representation of the linkage map was initially produced with genetic_mapper.pl and edited for visual clarity in Adobe Illustrator (*Bekaert, 2016*). The breeding cross implemented did not allow for mapping of Y-chromosome markers because only one male was used to initiate the crosses.

## RESULTS

### Breeding results

One female from the Florida colony was crossed with one male from the Kansas colony. Twelve sibling pairs from the $F_1$ offspring were mated together and the pair that produced the largest number of $F_2$ progeny was chosen for sequencing. A total of 108 $F_2$ individuals and the parents and grandparents were used for further analysis.

### Genome size estimation

The genome size of *S. zeamais* was estimated by comparing the relative florescence of stained neural tissue from an unknown strain of *D. melanogaster* to stained neural tissue

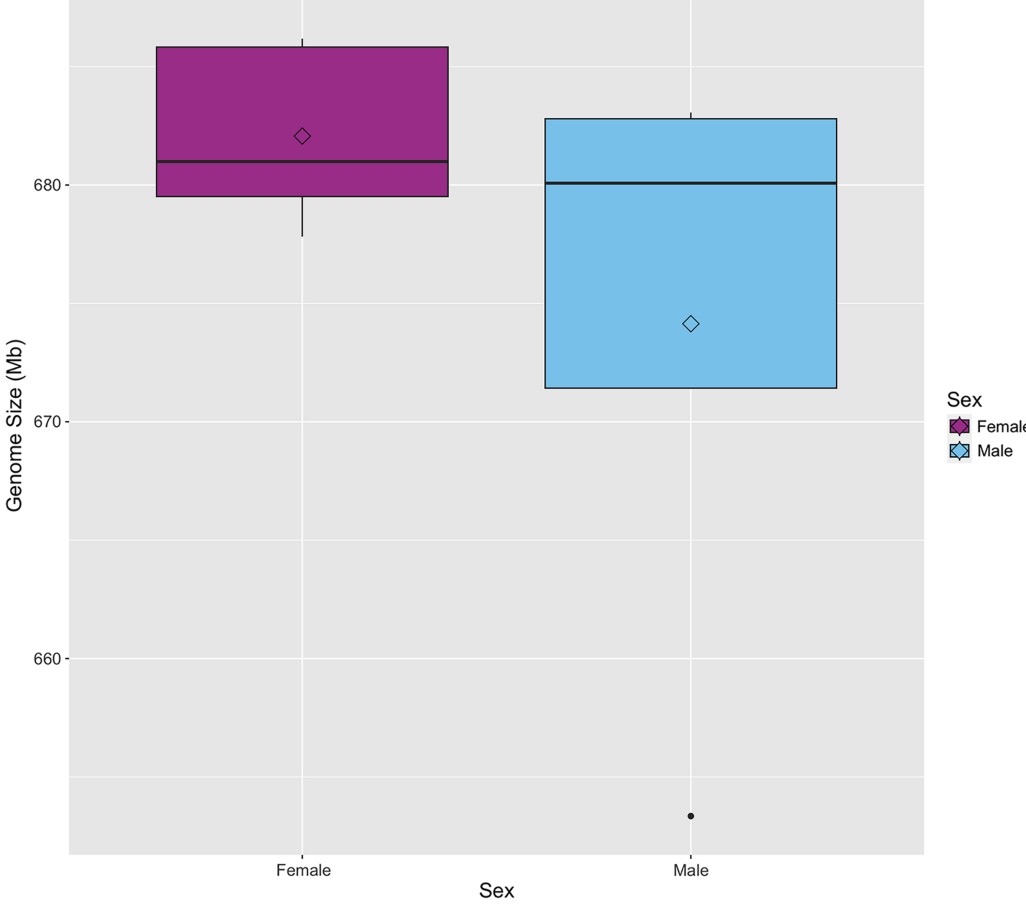

**Figure 1 Boxplot of genome size for male and female *Sitophilus zeamais*.** The mean, represented by the open diamonds, is 682 Mb (s.d. = 3.8) and 674 Mb (s.d. = 14.1) for females and males, respectively. The median for females is 681 Mb and the median for males is 680 Mb.

from *S. zeamais* (Table 1). Genome size was estimated for males and females separately since the species is heterogametic. The meioformula has been reported as $n = 10 + XX$ for females and $n = 10 + Xyp$ for males (*da Silva et al., 2015*, *2018*). A total of five females were measured and the average haploid genome size was 682 Mb (SE +/− 1.7). Additionally, five males were measured, but only four were used to estimate the genome size because the *D. melanogaster* control for one individual failed to stain and fluoresce. Males had an average genome size of 674 Mb (SD +/− 7.0). The medians were 681 Mb and 680 Mb for females and males, respectively (Fig. 1). We note that B chromosomes vary among individuals of the species, and may confound size estimates of the standard karyotype.

## Linkage map

The map was produced using ddRadSeq markers (radtags) from the F1 parents and 104 of the 108 $F_2$ progeny because four individuals failed to produce an adequate number of sequencing reads. The average number of reads per progeny used in the analysis was 914,379. The average number of reads of the four excluded samples was 5,910. The lowest

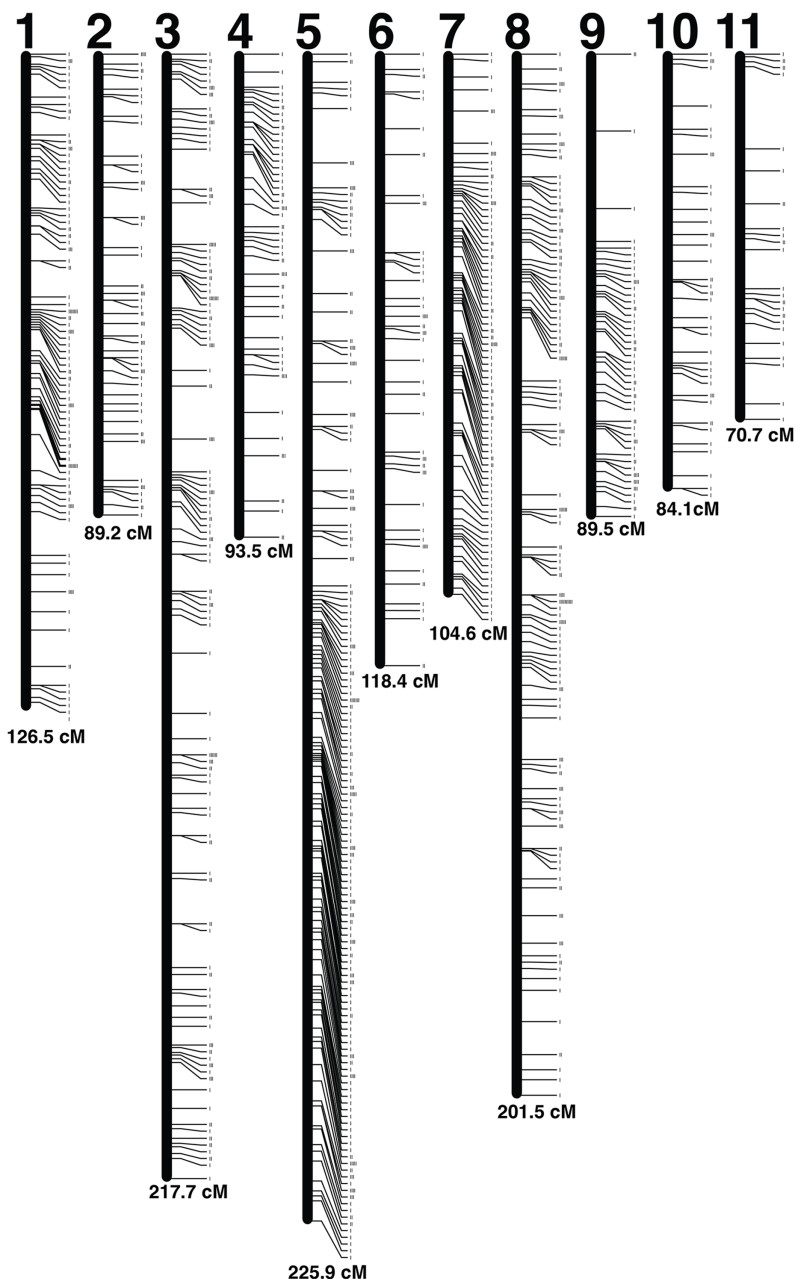

**Figure 2 Linkage map.** Each linkage group is represented by a black bar and is numbered at the top. The distance (cM) for each linkage group is listed at the bottom and each marker is represented by a tick mark to the right side of the linkage groups.

number of reads per individual for the retained samples was 315,155. The number of reads for the male and female parents were 965,533 and 960,524 respectively.

A set of 5,992 radtags were exported from the Stacks *populations* pipeline and filtered based on Mendelian inheritance and number of individuals genotyped per radtag. Radtags that did not meet Mendelian inheritance based on a chi-squared test and radtags that were not present in 100 out of 104 offspring were removed. A final set of 1,123 radtags with sequence length of 116 bp was used to produce the linkage map (Fig. 2). The map was

**Table 1 Flow cytometry data.** Sex of maize weevil, relative fluorescence of the standard (female *Drosophila melanogaster*), and the relative fluorescence of the experimental *Sitophilus zeamais* are recorded.

| Sample | Sex | Fluorescence$_{Dm}$ | Fluorescence$_{Sz}$ |
|--------|--------|--------|--------|
| 1 | Female | 22,617 | 88,682 |
| 2 | Female | 22,233 | 86,517 |
| 3 | Female | 22,521 | 87,448 |
| 4 | Female | 22,508 | 88,209 |
| 5 | Female | 22,769 | 88,190 |
| 6 | Male | 21,904 | 85,452 |
| 7 | Male | 22,720 | 84,823 |
| 8 | Male | 22,013 | 85,215 |
| 9 | Male | 22,203 | 86,663 |

**Table 2 Linkage map summary statistics.** A table showing the number of markers, the average distance between markers in cM, and the total length in cM of each linkage group.

| Linkage group | Num markers | Length (cM) | Mean distance (cM) |
|--------|--------|--------|--------|
| 1 | 119 | 126.47 | 1.06 |
| 2 | 68 | 89.21 | 1.31 |
| 3 | 162 | 217.67 | 1.34 |
| 4 | 65 | 93.5 | 1.44 |
| 5 | 235 | 225.87 | 0.96 |
| 6 | 59 | 118.38 | 2.01 |
| 7 | 99 | 104.58 | 1.06 |
| 8 | 173 | 201.54 | 1.16 |
| 9 | 66 | 89.49 | 1.36 |
| 10 | 46 | 84.07 | 1.83 |
| 11 | 29 | 70.69 | 2.44 |

made using a LOD score of 11, which generated 11 linkage groups using 1,121 radtags (Table S1). Two radtags did not map to any linkage group. The total map length was 1,421.47 cM and linkage groups ranged in size from 70.69 cM to 225.87 cM (Table 2). There were an average of 101.9 markers per linkage group (range: 29–235) with an average spacing of 1.45 cM between markers. The linkage map with LOD score of 11 was chosen because it produced the map with the highest LOD score that contained 11 chromosomes, which best corresponds to the karyotype of 10 autosomes and 1 X-chromosome (*Smith & Brower, 1974*; *Zhi-Yua, Pei & Guo-Xiong, 1989*; *da Silva et al., 2015*, *2018*). By lowering the LOD threshold to six all 1,123 radtags could be incorporated into a map. That map contained 11 linkage groups and had a total map length of 1,470.53 cM. But, this map was not chosen as the lower LOD score provides less confidence in the overall linkage map.

## DISCUSSION

*S. zeamais* is a worldwide pest that disproportionately affects subsistence farmers in developing countries. Damage from this pest threatens food security in these communities as widely available and effective control methods are lacking. With the revolution of genetic pest management, the potential to control this species at an ecosystem scale, which could reduce the cost burden on farmers, may be possible (*Esvelt et al., 2014*; *Baltzegar et al., 2018*). However, achieving this technical goal requires sophisticated genomic tools, very few of which currently exist for *S. zeamais*. Here, we have measured the genome size and developed the first genetic linkage map for this species.

Knowledge of an organism's genome size is a necessary parameter for many sequencing projects because this value allows for proper experimental design to obtain sufficient sequencing coverage. The haploid genome size was estimated using the gold standard method of comparing relative fluorescence measured *via* flow cytometry as 682 Mb and 674 Mb for females and males, respectively. This is somewhat smaller than the 713.5 Mb (females) and 709.3 Mb (males) previously reported (*da Silva et al., 2018*). Genome size varies naturally within different lines of *D. melanogaster* (*Huang et al., 2014*). Therefore, the difference in genome size here compared to the previous study may be accounted for by a difference in the accuracy of the estimated genome size used for controls. Also, the previous study used *Drosophila virilis* instead of *D. melanogaster* as a reference (*da Silva et al., 2018*). *D. melanogaster* has a smaller genome size than *D. virilis*. Its use as a standard in this study may have contributed to an underestimate in the genome size of *S. zeamais* due to the greater difference between the control and experimental samples and the nonlinearity inherent in flow cytometry analysis. Another possibility is that there exists a difference in the number of B chromosomes between sampled individuals in this study as compared to the previous. Supernumerary chromosomes are known to vary between populations as well as among individuals in *S. zeamais* (*da Silva et al., 2015*). The status of B chromosomes in the individuals used in this study is unknown. There were no B chromosomes in the line used for the original study (*da Silva et al., 2018*). Therefore, the difference in B chromosome presence is unlikely to account for the smaller size measured here. Future studies may consider using an average of the published genome sizes, especially if the status of B chromosomes in the population is unknown.

The linkage map contains 1,121 radtags segregating in a Mendelian fashion between two laboratory *S. zeamais* colonies. The map was resolved into 11 linkage groups, which correspond to the 10 autosomes and 1 X-chromosome found in the species. Future work should examine the Y-chromosome as some promising genetic pest management strategies rely on creating a male-biased population.

Although basic molecular tools exist for *S. zeamais*, the creation of a linkage map is an advancement in the development of modern genomic tools for the species. The linkage map will be a useful tool when assembling a genome. And, a completely sequenced and assembled genome will aid researchers by expanding the types of genetic manipulation and research questions that may be addressed. This linkage map also provides the necessary genome-wide sequence information required to perform quantitative trait loci (QTL)

analyses and genome-wide association studies (GWAS). Both QTL and GWAS could be used to identify useful phenotypic traits that may be targeted for future pest control methods.

## CONCLUSIONS

Development of genomic tools for *S. zeamais* may aid researchers in developing ecosystem wide genetic-based management solutions. This study measures the genome size and creates the first genetic linkage map for this species. The linkage map will be useful for assembling a complete genome sequence for this pest species and for providing the first set of genome-wide sequence data for QTL and GWAS studies.

## ACKNOWLEDGEMENTS

We would like to thank Nicole Gutzmann Martin, Destiny Tyson, and Laura Welsh for valuable laboratory assistance.

### Funding

This work was supported by a National Science Foundation IGERT award (No. 1068676). The funders had no role in study design, data collection and analysis, decision to publish, or preparation of the manuscript.

### Grant Disclosures

The following grant information was disclosed by the authors:
National Science Foundation IGERT award: 1068676.

### Competing Interests

The authors declare that they have no competing interests.

### Author Contributions

- Jennifer Fountain Baltzegar conceived and designed the experiments, performed the experiments, analyzed the data, prepared figures and/or tables, authored or reviewed drafts of the article, and approved the final draft.
- Fred Gould conceived and designed the experiments, authored or reviewed drafts of the article, and approved the final draft.

### Data Availability

The sequence data is available at Zenodo and BioProject: PRJNA910170; Baltzegar, Jennifer, & Gould, Fred. (2022). ddRadSeq sequences for *Sitophilus zeamais* [Data set]. Zenodo. https://doi.org/10.5281/zenodo.7087665

The code is available at Zenodo: Jennifer Baltzegar. (2022). jenbaltzegar/Sz_linkagemap: v1.0.0 (v1.0.0). Zenodo. https://doi.org/10.5281/zenodo.7086594

## Supplemental Information

Supplemental information for this article can be found online at http://dx.doi.org/10.7717/peerj.15414#supplemental-information.

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
