# Peer review of "Development of the first high-density linkage map in the maize weevil, Sitophilus zeamais"

_PeerJ, doi:10.7717/peerj.15414_

## Round 0.1 · original submission · Minor Revisions

The authors need to include reviewers' suggestions before re-submission of revised version.

Reviewer 1 ·

Basic reporting

See below

Experimental design

See below

Validity of the findings

See below

Additional comments

Development of the First High-Density Linkage Map in the Maize Weevil, Sitophilus zeamais
Jennifer Fountain Baltzegar1 and Fred L. Gould1,2



Line 36, what species? You list the articles but give me a few examples. Also what are the management techniques?

Line 42, add… feasible option for controlling the maize weevil..

Line 43…revise to… few genomic tools are available for use with the maize weevil…. To do what?

Objectives are clear

Line 109, what is “sib” mating?

Line 180, what are “inter se” crosses—I know what they are but be clear here.

Study is fairly clear, methods seem appropriate, conclusions also clear

Results valuable to the journal

avoid jargon or abbreviations without clearly defining them previously, as not everyone reading your work is an expert in this area

Line 26 please include full description the first time Sitophilus zeamais Motschulsky
Line 39 please use S. zeamais throughout, not common name.

Reviewer 2 ·

Basic reporting

Development of the first high-density linkage map in the maize weevil, [i]Sitophilus zeamais

The authors review well the current information on the maize weevil. The paper is well organized and easy to read and understand.
The data presented include the genome size and the rad sequencing linkage mapping.
The genome size was precisely determined in an earlier cited publication. It is good to have the estimate verified and the paper does that. The genome size reported is smaller than the earlier published value and the explanations given for the lower value are likely incomplete. It is possible that the measured maize weevil strain has a lower genome size, but unlikely that biological variation accounts for all the difference. The likely reasons the new estimate is low is: 1) the use of an unspecified D. melanogaster strain with an assumed genome size of 175Mbp. Huang et al. 2014 and other publications have documented considerable genome size variation among D. melanogaster strains, with the genome size of many strains larger than 175Mbp and 2) the use of a D. melanogaster standard, rather than a standard closer in size to the maize weevil will also underestimate the correct genome size due to well documented nonlinearity typical of most flow cytometers. The original estimate was based on a verified D. virilis standard that is nearly twice the genome size of the yw D. melanogaster strain whose genome size is 175 Mbp Additionally, the next-gen flow cytometer used for the original estimate is exceptionally linear. It would be good to see some .recommendation for readers whether the authors support the original estimate or in the future will use the average of the original and verified value.
The RADTAG METHODS ARE REALY WELL WRITTEN. This will be a boon for others doing RADTAG. However, it is not clear to this reviewer what the reader will learn regarding the linkage map itself. Unless the map is a available with the SNPs described at each of the 1121 locations, the data is useless. At the least, the paper might mention whether the data is available on request.
The markers are SNPs. This should be clear when markers are first mentioned. SNP appears later at present.
On first read, it is not clear how the SNPs are reduced from thousands to 1223 and finally to 1121. The reader should not be expected to know what the program does. The current statement regarding the reduction is vague and uninformative.
There is a good deal said about the importance of molecular data on S. zeamaize. However this is non-technical and is like “preaching to the choir.” Those who will use the molecular data will learn nothing new from what is written here. What is missing is a more detailed explanation of why the data presented here will be valuable to the readers. My impression is that the map data will be of value only to the authors - unless the data is made public. The only stated reason for the data is to produce a complete S. Zeamaize genome. Is that the message the author want to send?
Last is the length of the total linkage group. Does the presence of two unmapped SNP suggest they are on a “B” chromosome or unlinked? Do the authors want to go on record with the linkage size given here. If not, a comment on the estimate relative to the completed linkage map will be useful. Last but certainly not least is
1) The reported average genome size is the Haploid (1C) genome. This should be clear. Because the value is an average, the standard error, rather than the Standard deviation is the appropriate measure of confidence.
2) The male female difference. Both the original and the estimate here show very little difference between the genome size of the male and female. Much can be gleaned from that. The first is very poorly addressed as the modal value rather than the average. The mode shows the male value is skewed. The skew is due to a single male with a smaller genome size. I suspect this is because the other males have B chromosomes. I say this, because the manuscript has what appears to be a major error regarding the sex determination system. The system is described as X/Y. It has been reported as X1X2/Y. Males and females may differ by little in a new X/Y system. It seems likely that the sexes will differ considerably for an X1X2/Y. The information regarding the sex determining system, and the implications with regard to the genome size estimates should be corrected and expanded.
Reference
Huang, W., A. Massouras, Y. Inoue, J. Peiffer, M. Rámia, A. Tarone, L. Turlapati, T. Zichner, D. Zhu, R. Lyman, M. Magwire, K. Blankenburg, M. A. Carbone, K. Chang, L. Ellis, S. Fernandez, Y. Han, G. Highnam, C. Hjelmen, J. Jack, M. Javaid, J. Jayaseelan, D. Kalra, S. Lee, L. Lewis, M. Munidasa, F. Ongeri, S. Patel, L. Perales, A. Perez, L. Pu, S. Rollmann, R. Ruth, N. Saada, C. Warner, A. Williams, Y-Q. Wu, A. Yamamoto, Y. Zhang, Y. Zhu, R. Anholt, J. Korbel, D. Mittelman, D. Muzny, R. Gibbs, A. Barbadilla, J. S. Johnston*, E. Stone, S. Richards*, B. Deplancke*, and T. Mackay*. (2014). Natural variation in genome architecture among 205 Drosophila melanogaster Genetic Reference Panel lines
Baxter S.W., J.W. Davey, J.S. Johnston, A.M. Shelton, D.G. Heckel, C.D. Jiggins, and M.L. Baxter. (2011). Linkage mapping and comparative genomics using next-generation RAD sequencing of a non-model organism

Experimental design

No comment, except to mention that the D. melanogaster strain should be given and the relatively small size of the standard relative to the sample likely contributes to an underestimate,

Validity of the findings

Except for the underestimate of the genome size, for the reasons given and the small number of SNP mapped which is glossed over, the findings are very likely sound.

My biggest concern is how the readers access the map data.

---

## Round 0.2 · accepted · Accept

The authors have revised the manuscript. The manuscript is now acceptable for publication.

Reviewer 2 ·

Basic reporting

the revised MS is much clearer. Ambiguities have been addressed. The literature references have been expanded -a significant improvement. References to tables and data availability is now clear and easy to follow.

Experimental design

The revised MS addresses questions about methods. They are now sufficient to allow replication.
Importantly, the analysis of radtags, rather than SNP's is now clear.

Validity of the findings

The error of reviewer two regarding sex-determination is addressed, although the questions regarding genome size differences between males and females and between the initial estimate and the estimates presented here remain.

Additional comments

An excellent revision. Thank you for the obvious effort made to address reviewer comments, including my error regarding the XpY sex determination system.